# SARS-CoV-2 Spike Protein (RBD) Subunit Adsorption at Abiotic Surfaces and Corona Formation at Polymer Particles

**DOI:** 10.3390/ijms232012374

**Published:** 2022-10-15

**Authors:** Paulina Żeliszewska, Monika Wasilewska, Piotr Batys, Katarzyna Pogoda, Piotr Deptuła, Robert Bucki, Zbigniew Adamczyk

**Affiliations:** 1J. Haber Institute of Catalysis and Surface Chemistry Polish Academy of Science, Niezapominajek 8, 30-239 Cracow, Poland; 2Institute of Nuclear Physics, Polish Academy of Sciences, 31-342 Kraków, Poland; 3Department of Medical Microbiology and Nanobiomedical Engineering, Medical University of Białystok, 15-222 Białystok, Poland

**Keywords:** adsorption of S protein unit, aggregates of S protein, corona of S protein, S protein corona, SARS-CoV-2 spike protein, zeta potential of S protein

## Abstract

The adsorption kinetics of the SARS-CoV-2 spike protein subunit with the receptor binding domain at abiotic surfaces was investigated. A combination of sensitive methods was used such as atomic force microscopy yielding a molecular resolution, a quartz microbalance, and optical waveguide lightmode spectroscopy. The two latter methods yielded in situ information about the protein adsorption kinetics under flow conditions. It was established that at pH 3.5–4 the protein adsorbed on mica and silica surfaces in the form of compact quasi-spherical aggregates with an average size of 14 nm. The maximum coverage of the layers was equal to 3 and 1 mg m^−2^ at pH 4 and 7.4, respectively. The experimental data were successfully interpreted in terms of theoretical results derived from modeling. The experiments performed for flat substrates were complemented by investigations of the protein corona formation at polymer particles carried out using in situ laser Doppler velocimetry technique. In this way, the zeta potential of the protein layers was acquired as a function of the coverage. Applying the electrokinetic model, these primary data were converted to the dependence of the subunit zeta potential on pH. It was shown that a complete acid-base characteristic of the layer can be acquired only using nanomolar quantities of the protein.

## 1. Introduction

Coronavirus attachment to abiotic surfaces (fomites) is a crucial issue controlling their inactivation and removal by filtration (various kinds of masks) which can decrease the proliferation rate of epidemics. However, despite its essential significance, few experimental works were devoted to a thorough physicochemical analysis of this process. This is mainly caused by limited availability of intact virus (virions) and highly restrictive procedures governing permission to undertake such research [1].

It is reasonable to expect that one can overcome this limitation considering that virus interaction with abiotic surfaces and receptors is mainly controlled by the transmembrane proteins, referred to as spike (S) proteins forming protective coronas [2,3,4,5,6]. For example, detailed information about the S-protein corona of the SARS-CoV-2 virus was given in Refs. [7,8,9,10,11,12,13,14]. In particular, molecular dynamics calculations performed in Ref. [8] showed that the protein molecule has the molar mass of 141 kg mol^−1^ and assumes a cube-like shape with the length 26 to 28 nm, for the closed and open conformation, respectively. The cross-section of the molecule resembles a triangle having the edge of 15 nm. Both the shape and the dimensions of the molecule were experimentally confirmed in Ref. [7] applying cryo-electron microscopy measurements. It was also established [13] that the upper part of the S-protein contains a domain responsible for the virion attachment to receptors, referred to as the receptor binding domain (RBD) [15]. It should be underlined that in contrast to whole S-protein, its subunits containing the RBD domain can be synthesized in recombinant form using conventional procedures [16,17], which opens the possibility for experimental investigations of its properties and attachment mechanism.

Unfortunately, few experimental works concerning the bulk physicochemical characteristics of the RBD containing subunits of SARS-CoV-2 virus and their interactions with abiotic surfaces have been reported in the literature. In Ref. [18], the kinetics of the S1 subunit adsorption on alumina and titania plates under diffusion conditions was studied by high speed atomic force microscopy (AFM). The process was quantified as the dependence of the fraction of the surface area covered by the protein (defined as the coverage) on the adsorption time. It is shown that the characteristic time of monolayer formation on titania for pure water and pH 7.5 was equal to ca. 200 s. For the NaCl concentration of 0.165 M, especially for the alumina substrate, the monolayer formation time increased to 4000 s. This effect, analogous to that reported in Ref. [19] for the human serum albumin adsorption on a silica substrate, can be attributed to the protein aggregation at such a large electrolyte concentration. However, these results could not be quantitatively interpreted because the charge and zeta potentials of the protein and the substrates were not known. One should also mention that because of the AFM tip convolution effects appearing at a large coverage range, a valid estimation of the adsorbed molecule size, shape and aggregation degree could not be acquired.

In Ref. [20], the adsorption kinetics of the S1 and S2 subunits on gold sensors modified by various self-assembling layers was investigated by the quartz microbalance (QCM) for various pHs. The coverage of protein layers calculated using the Sauerbrey equation assumed the largest value of 7.3 mg m^−2^ at pH 7.4. A quantitative interpretation of these results was not attempted because of the lack of information about the molecule size and the zeta potential of the QCM sensors.

Given the deficit of reliable experimental data, the main goal of this work was to acquire physicochemical characteristics of the RBD containing subunit of the SARS-CoV-2 protein (referred later on as S1R) comprising aggregation degree, size and electrokinetic properties. A combination of sensitive methods was used such as atomic force microscopy yielding a molecular resolution, QCM and optical waveguide light-mode spectroscopy. The two latter methods furnish in situ information about the protein adsorption kinetics under flow conditions. The experiments performed for flat substrates such as mica and silica were complemented by investigations of the protein corona formation on polymer particles carried out using in situ electrokinetic techniques. Interpretation of the experimental data was carried out using the theoretical results derived from modeling, which yielded the charge of the protein as a function of pH. It is expected that the acquired information can enable development of efficient procedures for preparing stable S1R layers of well-defined coverage on solid substrates, which can be used in further studies for determining their interactions with macro-ion ligands, especially immunoglobulins.

## 2. Results and Discussion

### 2.1. Theoretical Charge Distribution and Hydrophobicity of the S1R Subunit

To facilitate the interpretation of experimental data, basic characteristic of the SARS-CoV-2 virus S protein and its subunits containing the receptor binding domain (RBD) were theoretically determined. The calculations were performed for using the protein crystallographic structure provided in Ref. [8]. In Figure 1, the secondary structures of the entire S protein, the S1 subunit (molar mass 74.6 kDa), the S1R subunit and the receptor binding motif (RBM), which is responsible for binding human angiotension-converting enzyme 2 (ACE2) are schematically shown.

The protonation states of the residues and the nominal charge of the molecule were calculated for a broad pH range using the PROPKA 3.0 algorithm [22,23,24]. The modeling performed in this work provided insight in the structure and shape of the molecule and determination of its nominal charge and hydrophobicity distributions, which are key factors in understanding its adsorption phenomena at abiotic surfaces. Primarily, from the amino-acid sequence it was determined that the molar mass of the S1R subunit in the monomeric state was equal to 25 and 25.8 kg mol^−1^ [kDa] for native and His-tagged version, respectively.

Using the typical value of the density of globular proteins, equal to 1.4 × 10^3^ kg mol^−1^ [25], one can calculate that the molecular volume is equal to 29.7 and 30.6 nm^3^ for the native and His-tagged molecule, respectively (see Table 1). This gives 3.84 and 3.88 nm, respectively, for the equivalent sphere diameter calculated as $d_{1}={\left(\frac{6v_{1}}{\pi }\right)}^{1/3}$. These values, which are practically equal within the bounds of calculation error, agree fairly well with the molecule size as approximated by a sphere of such a diameter, see Figure 2c.

It is also established in the calculations that the nominal charge of the native S1R unit remains positive for the entire pH range 1 to 9, and equal to 23 and 7.5 *e* (where *e* is the elementary charge assumed to be positive) at pH 3, and 7.0, respectively (see Figure 3). As can be expected, the nominal charge of the recombinant protein is slightly larger in the entire pH range, in comparison to the native protein, due to the positive charge of the attached His-tag, see Figure 3.

It should be noted, however, that the nominal charge in electrolyte solutions is largely screened by counter-ion condensation. Thus, under physiological conditions characterized by 0.15 M NaCl concentration, the screening length is below a nanometer and in effect the effective molecule charge is only a small fraction of the nominal charge as discussed in Refs. [25,27]. Under such conditions the protein molecules may show the tendency to aggregation, promoted by an uneven charge distribution and the hydrophobic amino-acid residues as shown in Figure 2a,b.

### 2.2. Characteristics of Substrates Used in Adsorption Experiments

Thorough physicochemical characterization of mica and silica sensors used in S1R adsorption experiments was carried out using AFM measurements (surface topography) and streaming potential measurements w carried out in the four-electrode microfluidic cell Ref. [28]. It was established that the root mean square (rms) parameter for mica sheets determined by AFM was equal to 0.1 ± 0.02 nm, whereas for the QCM sensors it was equal to 0.8 ± 0.2 nm. The dependencies of the mica and silica zeta potential on pH derived from the streaming potential measurements are shown in Figure 4.

Analogous bulk characteristics of the polymer particles used as carriers in the corona formation experiments were performed using DLS (diffusion coefficient) and the LDV measurements (electrophoretic mobility). Using these experimental data, the hydrodynamic diameter of particles was calculated using the Stokes–Einstein formula and the zeta potential using the Smoluchowski–Henry relationship. The hydrodynamic diameter of the particles derived in this way was equal to 820 ± 20 and 800 ± 20 nm for the 10 and 150 NaCl concentration, respectively, which agreed within error boundaries with data derived from the diffractometer. The zeta potential of the S800 particles derived via electrophoretic mobility measurements was equal to −97 and −105 mV at pH 3.5 and 7.4 (for the ionic strength 0.01 M). The dependence of the zeta potential of particles on pH for 0.01 and 0.15 M ionic strength is shown in Figure 4.

### 2.3. S1R Adsorption: AFM, QCM and OWLS Measurements

Adsorption of the S1R molecules on mica under diffusion transport was determined by AFM according to the procedure previously applied for fibrinogen [29] and human serum albumin [19]. Because of the molecularly smooth and exceptionally homogeneous surface properties of mica, this method allowed us to determine the dimensions, shape and the number of protein molecules per unit area of the interface (referred to as the surface concentration). Measurements were carried out at pH 3.5, 5.6 and 7.4 (PBS). Typical AFM micrographs of the protein layers acquired for the adsorption times of 15, 30 and 60 min (pH 5.6) are shown in Figure 5. As can be seen, for the short adsorption time, the aggregates were separated from each other by an average distance larger than their dimensions that allowed us to precisely determine their size, and surface concentration.

A qualitative analysis of these micrographs indicated that the aggregatess were quasi-spherically shaped. Quantitatively, the size distribution of aggregates was determined by measuring their dimensions in two perpendicular directions and taking an average value. The size histogram obtained in this way is shown in Figure 6. It was determined that the average size of the aggregates was equal to 14 ± 4 nm. On the other hand, analogous AFM investigations performed at pH 7.4 (PBS buffer) showed that the average aggregates size was equal to 18 ± 5 nm.

Thus, from the AFM measurements it becomes evident that the average particle size significantly exceeds the S1R monomer size predicted to be 3.84–3.88 nm. This indicates that the protein molecules were aggregated to a considerable degree. The geometrical size of various aggregates schematically shown in Table 2 can be predicted considering the above monomer size, assuming its quasi-spherical shape. Using the geometrical size one can calculate the AFM size of the aggregate as an average from two different orientations. The hydrodynamic diameter of the aggregates up to the tetramer was calculated using the results presented in Ref. [30]. For larger aggregates the approximate formula derived in Ref. [30] was used, assuming their quasi-spherical structure (see Table 2). Thus, the predicted AFM size of the dimer is equal to 5.8 nm, which agrees with the smallest size in the histogram shown in Figure 6. Analogously, the AFM sizes of the aggregate composed of 12 monomers is equal 12 nm. Finally, the experimental value of 14 nm corresponds to the S1R aggregate composed of 20 monomers forming a micelle of a spherical shape.

It should be mentioned that a spontaneous formation of S1R aggregates indicates that the molecule charge and hydrophobicity are heterogeneously distributed as was predicted from the theoretical modeling. In consequence, the more hydrophobic and less charged part of monomers can attract each other whereas the charged and hydrophilic parts are exposed to the electrolyte solution, which was previously observed for the vimentin molecule [30].

Except for the size, one can also calculate the effective cross-section area of the aggregates adsorbed on a surface under the side-on and end-on orientations. They varied between 24 and 100 nm^2^ for the dimer and the 20 molecule aggregate (for the side-on orientation). Using these data one can calculate the maximum coverage of aggregates, expressed as a protein mass per unit area and denoted by *Γ*, from the formula:(1)$$\Gamma =\Theta_{mx}\left(\frac{n_{a}M_{1}}{S_{g}N_{Av}}\right)$$
where $\Theta_{mx}$ is the dimensionless maximum coverage, which for non-interacting spherical particles is equal to 0.55 [31], $n_{a}$ is the number of monomers in the aggregate. It is calculated from Equation (1) that *Γ*_∞_ varies for the side-on orientation between 1.6 and 3.4 mg m^−2^ for the dimer and the 20 molecule aggregate, respectively (see Table 2).

**Table 2 ijms-23-12374-t002:** Predicted physicochemical characteristics of S1R protein aggregates.

Aggregate/Parameter	*d_H_**d_AFM_*[nm]	*S_g_**_‖_*[nm^2^]	*S_g_*_┴_[nm^2^]	*Γ*_∞‖_[mg m^−2^]	*Γ*_∞┴_[mg m^−2^]
Monomer, *n_a_* = 1*M*_1_ = 22 kDa* 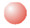 *	3.84				
3.84	12	12	1.6	1.6

Dimer, *n_a_* = 2*M*_2_ = 44 kDa 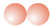	5.45.8	24	12	1.6	3.2

Tetramer, *n_a_* = 4*M*_4_ = 88 kDa* 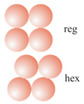 *					

7.69.2	52	24	1.5	3.6

Dodekamer(micelle), *n_a_* = 12*M*_12_ = 264 kDa 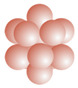	1212	87	87	2.6	2.6
Micelle, *n_a_* = 20*M*_20_ = 440 kDa 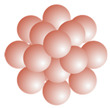					
1414	100	100	3.4	3.4

*d_H_*—hydrodynamic diameter of aggregates, calculated numerically for *n_a_* = 2, 4 [30] and from the formula $d_{H}=C_{H}n_{a}^{1/3}d_{1}$ (*C_H_* = 1.35) for *n_a_* > 10 (*n_a_*—aggregation number), *d_AFM_*- hydrodynamic diameter for AFM measurements, *S_g_**_‖_* cross-section area for the side-on adsorption, *S_g┴_* cross-section area for the end-on adsorption, *Γ*_∞‖_, maximum coverage for the side-on adsorption, *Γ*_∞┴_, maximum coverage for the end-on adsorption, calculated from Equation (1).

Measurements of the adsorption kinetics of S1R on silica sensors were carried out using a quartz crystal microbalance (QCM) according to the procedure previously applied for other proteins [19,32]. Although this method is precise enabling real-time, in situ measurements of adsorption/desorption kinetics under convection conditions, the interpretation of the results is often not unique. This is because the primary signals acquired in the QCM measurements, i.e., the oscillation frequency and the dissipation shifts, depend on the force exerted on the sensor rather than on the particle mass [32]. The force has an inertia component proportional to the particle mass and the hydrodynamic component, which plays a significant role for small particle size and lower overtone numbers. This hydrodynamic component is often interpreted as the solvent coupling effect leading to apparent hydration of particle or protein layers. Therefore, the significance of the results derived from QCM can be increased if complementary measurements are carried out under similar physicochemical conditions using an experimental method, which could yield the absolute (dry) coverage of the protein. In this work we used the reliable OWLS method, whose validity for investigations of globular protein adsorption was previously confirmed [19].

Typical QCM and OWLS kinetic runs performed at pH 3.5 and 10 mM NaCl concentration are shown in Figure 7. As can be seen, in the case of QCM measurement, the protein coverage (calculated using the Sauerbrey equation for the overtones 3, 5, 7, 9, 11) rapidly increases attaining a maximum value of 8.5 mg m^−2^ (for the 3rd overtone) after 40 min of adsorption. Under the desorption run, where pure electrolyte of the same ionic strength and pH was flushed through the cell with the same flow rate, the protein coverage slightly decreased and attained a stationary value of 8.0 mg m^−2^ (for the 3rd overtone). The stationary coverage monotonically decreased with the overtone number attaining 6.5 mg m^−2^ for the 11th overtone.

It is interesting to mention that in the case of fibrinogen (molar mass equal to 340 kg mol^−1^) the maximum QCM coverage after desorption (calculated for the 3rd overtone) was equal to 3.0 mg m^−2^ at pH 3.5 and 10 mM NaCl [32]. Considering the hydrodynamic coupling effect, the dry coverage of fibrinogen equal to 1.2 mg m^−2^ was determined, which yielded the value of 0.6 for the apparent hydration function. Interestingly, a similar value of the hydration function of 0.55 was obtained for the human serum albumin [19]. Taking this hydration function one can calculate that the dry coverage of S1R varied between 3.2 and 2.6 mg m^−2^ for the 3rd and 11th overtones, respectively. One can notice that these values agree fairly well with those theoretically predicted for the aggregates composed of 20 and 12 monomers, see Table 2. The validity of these maximum coverage calculations is confirmed by the OWLS adsorption kinetic measurements of S1R shown in Figure 7, which is analogous to that of QCM; i.e., the protein coverage increased linearly with time and attained a plateau value of 2.8 mg m^−2^ after 40 min. Upon switching to the desorption run initiated at 110 min, the coverage slightly decreased to 2.6 mg m^−2^, which agrees with the above QCM results for the 11th overtone.

Thus, the results shown in Figure 7 confirm an irreversible adsorption mechanism of S1R, which can interpreted in terms of attractive electrostatic interactions given that at pH of 3.5 the charge of the molecules predicted from the modeling (see Figure 3) was positive, whereas the silica charge (zeta potential) was negative at this pH (see Figure 4a).

Analogous adsorption/desorption kinetic experiments were performed at pH 7.4 fixed by the PBS buffer at ionic strength of 10 and 150 mM. The results shown in Figure 8 indicate that the QCM coverage attained after the desorption run was equal to 3.0 and 2.0 mg m^−2^ (3rd overtone) for the 150 and 10 mM NaCl concentrations, respectively, which corresponds to the dry coverage of 1.2 and 0.8 mg m^−2^ calculated using the hydration function pertinent to fibrinogen. These values agree with the maximum dry coverage derived from OWLS (see Figure 8b), which supports the applicability of the QCM measurements. The significant decrease in the coverage at pH 7.4 compared with pH 3.5 can probably be explained by the decrease in the charge of the His-tagged S1R molecule predicted from the modeling, which was equal to 26 and 9 e, at pH 3.5 and 7.4, respectively.

### 2.4. S1R Protein Corona Formation on Polymer Particles

As demonstrated in previous works [33,34], investigations of protein adsorption on carrier particles exhibit numerous advantages compared to the planar substrate adsorption discussed above. Primarily, the monolayer formation time is considerably shorter, typically of the order of a second and it does not depend on the bulk protein concentration [34]. Therefore, experiments can be efficiently carried out even for nM solutions, which is especially important for expensive proteins such as the virus S proteins and immunoglobulins. The density of the protein corona formed in this way, mimicking in principle, inactive virion structure [35,36], can be controlled in situ using electrophoretic mobility measurements carried out using the LDV method.

In this work we applied this method to investigate the S1R corona formation on the negatively charged polystyrene particles (S800). Primary experimental results showing the dependence of the particle zeta potential on the initial concentration of the S1R protein acquired at pH 3.5 and 10 mM NaCl are plotted in Figure 9.

As can be observed, the initially negative zeta potential of particles rapidly increases with the protein concentration and changes its sign to positive at *c_b_* = 1 mg L^−1^. This corresponds to the S1R aggregates concentration of 2.2 nM assuming its molar mass of 440 kDa. Given that zeta potential can be effectively determined using the LDV measurements, with a relative precision of 5% for this range, this result indicates that using the LDV method, the S1R concentration can be determined below one nM. However, for the protein coverage larger than 1 mg L^−1^ the increase in the zeta potential becomes less steep and finally a plateau value of 23 mV is attained.

The experimental results shown in Figure 9 were interpreted using the general electrokinetic model discussed in Ref. [37] where the following expression for the zeta potential of protein corona at carrier particles was formulated
(2)$$\zeta (\Theta )=F_{i}(\Theta )\zeta_{i}+F_{p}(\Theta )\zeta_{p}$$
where *ζ*(*Θ*) is the zeta potential of the corona, *ζ_i_* is the zeta potential of bare particles, *ζ_p_* is the protein zeta potential in the bulk, and *F_i_*(*Θ*), *F_p_*(*Θ*) are the dimensionless functions. The *F_i_* function describes the damping of the local flow by the adsorbed molecule layer and the *F_p_* function characterizes the contribution to the zeta potential stemming from the electric double-layers around adsorbed molecules. The absolute (dimensionless) coverage in Equation (2) is calculated as
(3)$$\Theta =S_{g}N=S_{g}\left(\frac{{Ν}_{Av}}{M_{w}}\right)\Gamma$$
where *N* is the surface concentration of the protein molecules at the particle and Γ is the nominal mass coverage of the molecules calculated from the dependence [34]
(4)$$\Gamma =\left(\frac{\rho_{p_{}^{}}d_{p}}{6}\right)\frac{c_{b}}{c_{p}}$$
where $\rho_{p}$ and $d_{p}$ are the polymer particle density and the diameter, respectively, and *c_p_* is the particle bulk concentration.

The results calculated from Equation (2) for the S1R aggregate composed of 20 molecules (having the molar mass of 440 kDa) are plotted as a solid line in Figure 9. As can be seen, they adequately reflect the experimental run indicating that the maximum coverage (where the experimental zeta potential attains a plateau value) calculated from Equation (3) was equal to 3.0 mg m^−2^. This result agrees with the above discussed derived from the QCM and OWLS kinetic methods, which confirms a similar adsorption mechanism of S1R at microparticles and planar surfaces (sensors).

Additionally, knowing the limiting value of the corona zeta potential one can calculate the zeta potential of the S1R aggregates in the bulk by expressing Equation (2) in the following form
(5)$$\zeta_{p}=[\zeta (\Theta )-F_{i}(\Theta )\zeta_{i}]/F_{p}(\Theta )$$

Considering that for large corona coverage the $F_{i}(\Theta )$ function exponentially vanishes and the function approaches 0.71 [37], one obtains from Equation (5), $\zeta_{p}$ = 33 mV at pH = 3.5. It is interesting to mention that this value coincides with that of the recombinant albumin determined in Ref. [38]. This result indicates that the corona formation experiments enable us to determine in a robust way the bulk zeta potential using a microgram quantity of a protein, whereas adequate bulk electrophoretic mobility measurements (LDV) require a few orders of magnitude larger quantities, typically 1 mg.

It is also interesting to mention that the zeta potential acquired via the corona formation experiments can be used to determine the electrokinetic charge of the S1R aggregates from the following dependence [30]
(6)$$Q_{c}=2\pi \varepsilon d_{H}\left(1+\frac{1}{2}\kappa d_{H}\right)\zeta_{p}$$
where *ε* is the electric permittivity of the electrolyte, *d_H_* is the hydrodynamic diameter of the aggregate (Table 2) and $\kappa^{-1}$ is the electric double-layer thickness.

Considering that for 10 mM NaCl solution $\kappa^{-1}$ = 3.1 nm, *d_H_* = 14 nm and *ζ_p_* = 33 mV, one obtains from Equation (6) that the average number of elementary charges per aggregate is equal to 40 *e*, which gives 2 *e* per one S1R monomer. This amounts to only 8% of the theoretically calculated nominal charge of S1R monomer equal to 26 *e* at this pH. This can be attributed to a significant compensation of the charge by counter-ions, a common effect observed for other proteins such as myoglobin, for example [25].

It was also determined in a further series of experiments that the particle suspensions with the S1R coronas were stable over times exceeding 24 h. This enabled measurement of their electrophoretic mobility as a function of pH and in consequence the calculation, using the Smoluchowski equation, of their zeta potential. Results of such measurements, performed for the corona adsorbed at pH 3.5 with the coverage equal to 3 mg m^−2^ are shown in Figure 10.

As can be seen, the zeta potential of the particles becomes negative at pH ca. 5 and afterward it abruptly attains a plateau value of −40 mV, which does not change for pH up to 9. It is also worth mentioning that the zeta potential of the S1R corona did not change upon pH cycling, which confirms an adequate stability of the particles.

## 3. Materials and Methods

The recombinant His-tagged SARS-CoV-2 S1R subunit (lyophilized from sterile 40% acetonitrile, 0.1% TFA, producer molar mass 25 kg mol^−1^) was purchased from Raybiotech (#230-01102-1000, Peachtree Corners, GA, USA). The stock protein solution was diluted to a concentration of 1–5 mg L^−1^ prior to each adsorption experiment with pH adjusted by the addition of HCl (3.5–4) or sterile phosphate buffer saline (PBS supplied by Merck Darmstadt, Germany). The solution ionic strength was regulated by the addition of NaCl.

Other chemical reagents such as sodium chloride, sodium hydroxide, hydrochloric acid, were obtained from Avantor Performance Materials Poland S.A. (formerly POCH S.A., Gliwice, Poland). Ultrapure water, was obtained using the Milli-Q Elix&Simplicity 185 purification system from Millipore.

Ruby muscovite mica supplied from Continental Trade was used as a substrate for S1R protein adsorption. The solid pieces of mica were freshly cleaved to thin sheets prior to every experiment.

Negatively charged sulfonate polystyrene microparticles (referred to as S800), used as carriers in the subunit corona formation experiments, were synthesized according to the Goodwin procedure [39] and thoroughly purified by membrane filtration.

The kinetics of the protein adsorption was investigated by QCM measurements carried out using a Q-Sense E1 system (QSense, Gothenburg, Sweden) according to the standard procedure described in Ref [19]. Briefly, the protein solution with the desired concentration (typically 5 mg L^−1^) was introduced into the QCM-D cell under a regulated volumetric flow rate. After completing the adsorption run, the sensor was rinsed by the pure electrolyte in order to remove the loosely bound molecules. The adsorbed protein mass per unit area, hereafter referred to as the QCM-D mass (coverage), was calculated from the Sauerbrey equation: [40].
(7)$$\Gamma_{Q}=-C_{Q}\frac{\Delta f}{n_{o}}$$
where $\Gamma_{Q}$ is the mass coverage, Δ*f* is the frequency change, *n_o_* is the overtone number and *C_Q_* is the mass (coverage) sensitivity constant equal to 0.177 mg m^−2^ Hz^−1^ for the 5 MHz AT-cut quartz sensor.

The molecule adsorption kinetics was also measured using optical waveguide light mode spectroscopy (OWLS) according to the procedure previously described in Ref. [19]. The OWLS 210 instrument (Microvacuum Ltd., Budapest, Hungary) was used. The apparatus is equipped with a laminar slit shear flow cell comprising a silica-coated waveguide (OW2400, Microvacuum). The adsorbing substrates were planar optical waveguides made of a glass substrate (refractive index 1.526) covered by a film of Si0.78Ti0.22O2 (thickness 170 nm, refractive index 1.8) and an additional layer (10 nm) of pure SiO2 according to the previous protocol. It is worth mentioning that in contrast to the QCM measurements, which yield the wet mass of the protein layer comprising the hydrodynamically coupled water, the OWLS yields absolute (dry) values of the protein coverage [41].

Freshly cleaved mica sheets were used in protein adsorption experiments carried out under diffusion-controlled transport in a thermostatted cell. After finishing the adsorption run, typically lasting 30–180 min, the mica sheets were rinsed with ultrapure water for 30 s. The micrographs of the protein molecules were acquired by ambient air AFM imaging using the NT-MDT Solver device with the SMENA–B scanning head. The measurements were performed under semi-contact mode using polysilicon cantilevers NSG-03 with resonance frequencies of 47–150 kHz, a tip radius of 10 nm, and a cone angle less than 20°. The images of adsorbed protein molecules were acquired within the scan range of 2.0 μm per 2.0 μm, over 10–20 randomly chosen areas over the mica sheet, which ensures a relative precision of the coverage determination of about 3%. All the images were flattened using an algorithm provided with the instrument.

Bulk characteristics of the polymer particles used as carriers in the corona formation experiments were performed using the dynamic light scattering providing the diffusion coefficient and the laser Doppler velocimetry (LDV) yielding the electrophoretic mobility, (Zetasizer Nano, Malvern Instruments, Malvern, UK).

These primary experimental data enabled us to calculate the hydrodynamic diameter of particles using the Stokes–Einstein formula and the zeta potential using the Smoluchowski–Henry relationship. Independently, the size of the particles was determined by laser diffractometry and AFM.

The electrokinetic characteristics of the substrate (mica sheets and silica plates) were acquired by the streaming potential measurements carried out according to the previously described procedure [29].

The experimental temperature was kept constant at 298 K.

## 4. Conclusions

AFM investigations of S1R subunit adsorption on mica under diffusion transport furnished reliable information about the mechanisms of this process and basis physicochemical properties of the molecules. It was confirmed that the protein layer consisted of quasi-spherical aggregates with the size equal to 14–18 nm (for pH range 3.5–7.4). Formation of aggregates was explained in terms of a heterogeneous distribution of electric charge and hydrophobicity, which was predicted from the theoretical modeling.

These predictions were consistent with QCM and OWLS adsorption kinetic measurements on silica sensors and corona formation on polymer particles investigated by the LDV method. It was established, in both cases, that the aggregates efficiently adsorbed at pH 3.5 forming compact layers of the maximum coverage equal to 3 mg m^−2^. At pH 7.4 and 150 mM NaCl, the maximum coverage of S1R decreased to ca 1 mg m^−2^, which was attributed to their lower positive charge.

It was also shown that a quantitative interpretation of S1R corona formation on polymer particles was feasible in terms of the electrokinetic model. This can be exploited to efficiently prepare protein coronas on polymer particles of a controlled coverage. Additionally, using the electokinetic model, we determined the bulk zeta potential and the electrokinetic charge of the S1R aggregates, which was not previously reported in the literature. Such important information is impractical to derive from the bulk electrophoretic mobility measurements, which require orders of magnitude larger aggregate concentration compared to that used in the corona formation experiments.

The results obtained in this work confirm that it is feasible to produce stable S1R coronas on polymer particles with controlled coverage and zeta potential using only nanomolar quantities of the protein. Such layers can be exploited for investigations of the protein interactions with macromolecule ligands, particularly with immunoglobulins.

## Figures and Tables

**Figure 1 ijms-23-12374-f001:**
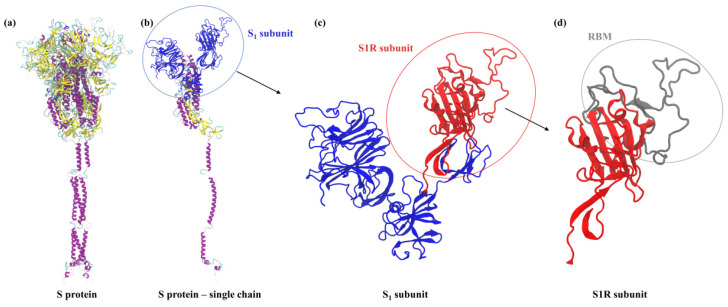
(**a**) Snapshot of S protein with highlighted secondary structure types. (**b**) Single chain of S protein with S1 subunit marked in blue. (**c**) The fragment of the S1 subunit with the receptor binding domain (S1R) marked in red. (**d**) Zoom of S1R domain with receptor binding motif (RBM) marked in grey. VMD software was used for visualization [21].

**Figure 2 ijms-23-12374-f002:**
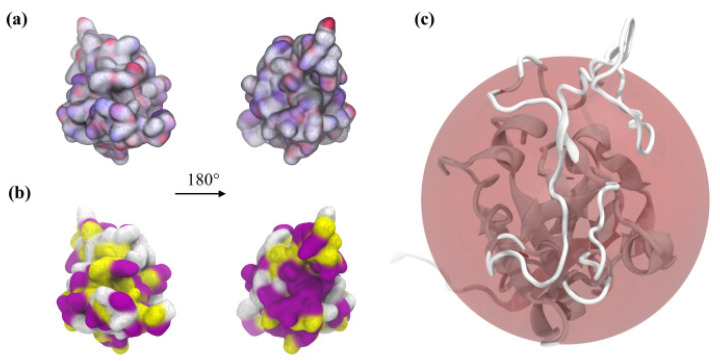
(**a**,**b**) correspond to the charge and polarity distribution of S1R domain, respectively. In (**a**), positive and negative charges are highlighted in blue and red, respectively. In (**b**), the polar and nonpolar amino acids are highlighted in violet and green, respectively. (**c**) S1R domain inscribed in the sphere of diameter 3.84 nm, see Table 2.

**Figure 3 ijms-23-12374-f003:**
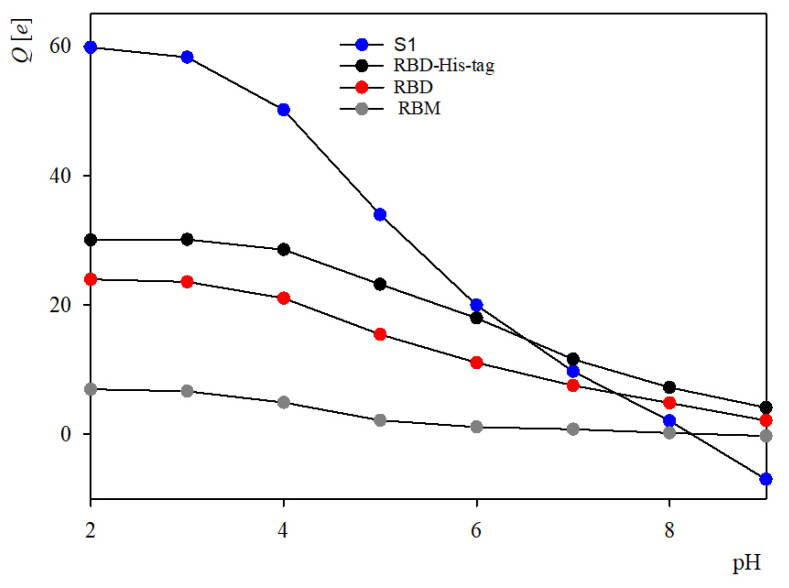
The nominal charge of the S1 subunit, the receptor binding domain, native and His-tagged, and the receptor binding motif (RBM) of SARS-CoV-2 S protein versus pH, determined using the PROPKA algorithm.

**Figure 4 ijms-23-12374-f004:**
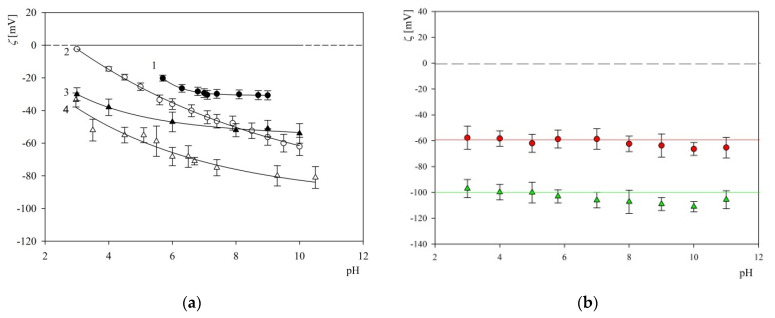
Part (**a**): The dependence of the zeta potential of mica and silica on pH determined by the streaming potential method: line 1—silica, 150 mM NaCl, line 2—silica, 10 mM NaCl, line 3—mica, 150 mM NaCl [28], line 4—mica, 10 mM NaCl. The solid lines represent fits of experimental data. Part (**b**): The dependence of the zeta potential of PS particles on pH determined by the LDV method. (●) 150 mM NaCl, (▲) 10 mM NaCl. The solid lines represent fits of experimental data.

**Figure 5 ijms-23-12374-f005:**
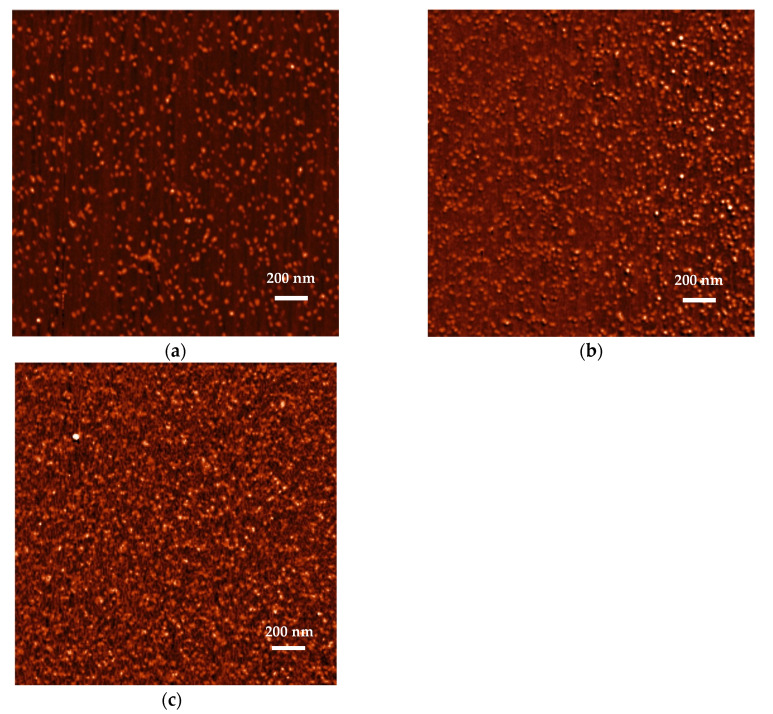
S1R protein layers on mica (AFM micrographs); adsorption conditions: pH 5.6, *c_b_* = 2 mg L^−1^, 150 M NaCl. Part (**a**): *t* = 15 min (protein surface concentration 250 μm^−2^), part (**b**): *t* = 30 min (protein surface concentration 350 μm^−2^); part (**c**): *t* = 60 min (protein surface concentration 450 μm^−2^).

**Figure 6 ijms-23-12374-f006:**
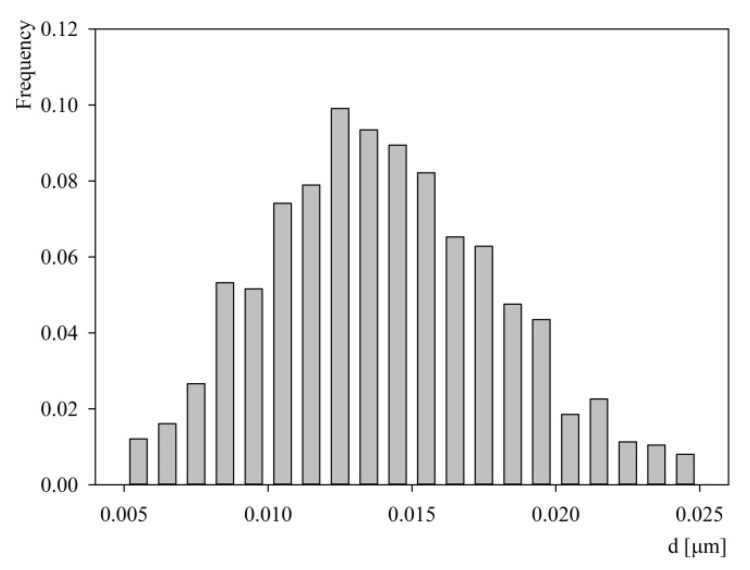
Histogram of the S1R protein size at mica determined by AFM; pH 5.6, *c_b_* = 2 mg L^−1^, 150 mM NaCl, adsorption time 15 min. Average aggregate size 14 ± 4 nm.

**Figure 7 ijms-23-12374-f007:**
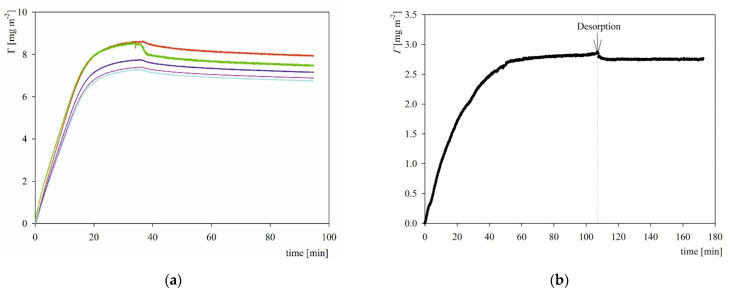
Kinetics of S1R adsorption/desorption expressed as the dependence of the coverage on the time. Experimental conditions: pH 3.5-4, electrolyte concentration 10 mM, bulk protein concentration 5 mg L^−1^ flow rate 8.3 × 10^−4^ cm^3^ s^−1^. Part (**a**) QCM measurements, silica sensor; the coverage was calculated using the Sauerbrey equation for the overtones 3 (red), 5 (green), 7 (blue), 9 (violet), 11 (light blue). Part (**b**) OWLS measurements, silica sensor.

**Figure 8 ijms-23-12374-f008:**
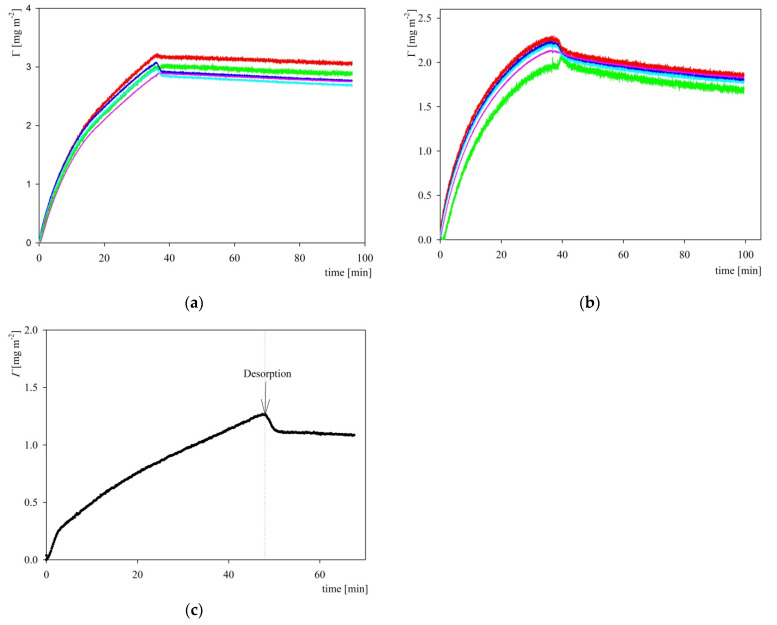
Kinetics of S1R adsorption/desorption expressed as the dependence of the coverage on the time; adsorption conditions pH 7.4 (PBS), bulk protein concentration 5 mg L^−1^, flow rate 8.3 × 10^−4^ cm^3^ s^−1^. Part (**a**) QCM measurements, silica sensor; 150 mM. Part (**b**) QCM measurements, silica sensor; 10 mM. The coverage was calculated using the Sauerbrey equation for the overtones 3 (red), 5 (green), 7 (blue), 9 (violet), 11 (light blue). Part (**c**) OWLS measurements, silica sensor, 10 mM, NaCl.

**Figure 9 ijms-23-12374-f009:**
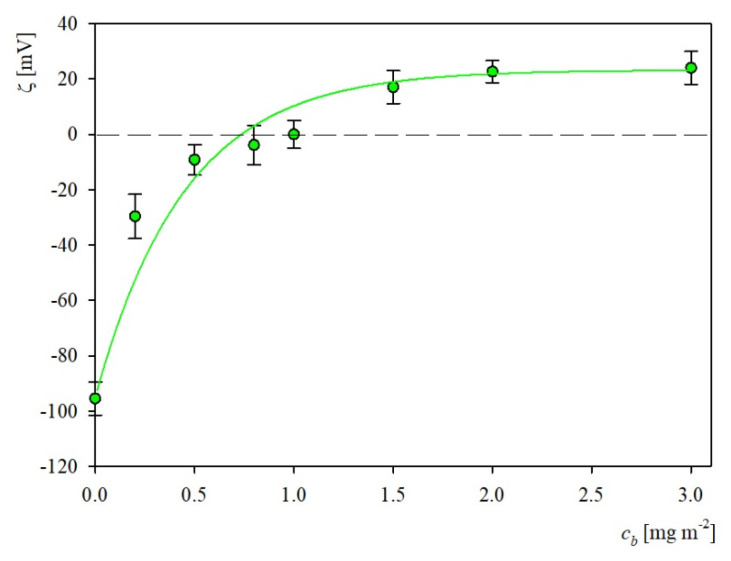
The dependence of the zeta potential of negatively charged polymer particles (S800) on the bulk S1R at protein concentration; adsorption conditions: pH 3.5, 10 mM NaCl, particle concentration 50 mg L^−1^. The points denote experimental results obtained from the LDV measurements; the solid lines show the theoretical results calculated from the electrokinetic model, Equation (2).

**Figure 10 ijms-23-12374-f010:**
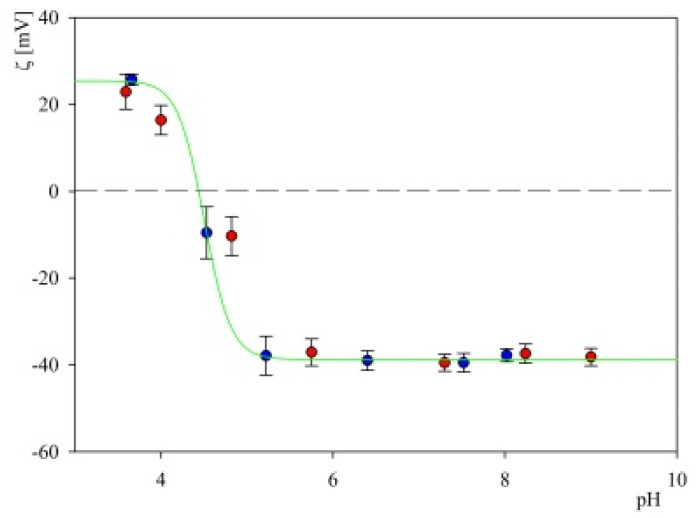
Dependence of the zeta potential of the S1R corona on S800 particles with the coverage equal to 3 mg m^−2^ on pH. The points represent experimental data acquired using the LDV method for 10 mM NaCl, (●)—pH increase; (●)—pH decrease, interpolated by a green line.

**Table 1 ijms-23-12374-t001:** Physicochemical parameters of SARS-CoV-2 S1R molecule monomer.

Property, Symbol Unit	Value	Remarks
Molar mass, monomer, *M*_1_ kg mol^−1^ [kDa]		
	25.0	Calculated from the sequence
	25.8 (His-tagged)	
Density, *ρ_p_* kg m^−3^	1.4 × 10^3^	assumed Ref. [26]

Specific volume, monomer, *v*_1_, nm^3^		
	29.7 30.6 (His-tagged)	Calculated as: $v_{1}=\frac{M_{1}}{\rho_{p}N_{Av}}$
Equivalent sphere (hydrodynamic) diameter, *d*_1_, nm	3.84 3.88 (His-tagged)	Calculated as: $d_{1}={\left(\frac{6v_{1}}{\pi }\right)}^{1/3}$
Geometrical cross-section area, of equivalent sphere, *S_g_*_1_, nm^2^	11.6 11.8 (His-tagged)	Calculated as: $S_{g1}=\frac{\pi d_{1}^{2}{}_{}}{4}$

$N_{Av}$—Avogadro number.

## Data Availability

The data is available on request.

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
