# Peer review of "SARS-CoV-2 Spike Protein (RBD) Subunit Adsorption at Abiotic Surfaces and Corona Formation at Polymer Particles"

_ijms, 2022, doi:10.3390/ijms232012374_

Round 1
Reviewer 1 Report
Reviewer’s comments on the manuscript: SARS-Cov-2 Spike Protein Subunit Adsorption at Abiotic Sur- 2 faces and Corona Formation at Polymer Particles written by P. Å»eliszewska, M. Wasilewsk, Piotr Batys, Zbigniew Adamczyk, Katarzyna Pogoda, Piotr DeptuÅ‚a and Robert Bucki
The reviewed manuscript presents the adsorption kinetic of SARS-Cov-2 spike protein subunit with the receptor binding domain at abiotic surfaces. The manuscript presents very interesting and worth reading results and in my opinion is in the journal’s fields of interests. However, I have same reservations thus my suggestion is major revision.
Reviewers comments and suggestions:
All manuscript: The manuscript is very carelessly prepared, which is surprising to me because the work was prepared by as many as seven authors and two of them are correspondent authors. Unfortunately, this significantly spoils the good impression of this work. It is good practice to read the final manuscript before submitting it to the editor. Some examples:
- lines 4, 5: names/initials and surnames of authors.
- line 18: 1 mg m-2.
- line 40: Molecular Dynamics – capital letters?
- lines 91-92: larger font?
- line 91: two parenthesis?
- Fig. 3: something like an unfinished frame of varying thickness. The same with Fig. 7b.
- line 218: editorial mistake.
- Fig. 4: why is the signature of the ordinates once horizontally and once vertically?
- Fig. 4. Unusual style of legend and figure caption. Please change them into the more conventional way.
- line 256: editorial mistake.
- lines 305-308: different colour and larger font, again.
- lines 327-328: why 3rd but 11th (not 11th)?
- line 329: editorial mistake.
- line 352: editorial mistake.
- line 355: editorial mistake.
- line 361: editorial mistake.
- line 379: editorial mistake.
lines 26-27: What is the difference between corona of S protein and S protein corona?
All manuscript: Sometimes there is a space between the number and unit sometimes there is not, please unify this.
All manuscript: number should be always in the same line that its unit. Please correct it in the whole manuscript.
All manuscript: This's probably the template's fault but often words are not divided correctly into syllables.
How can you explain only the slight dependence of the silica/150 mM NaCl system on changing pH (Fig. 4a)?
Fig. 5: Is it possible to present the AFM data in the more convincing way? It is really hard to analyze them.
line 425: Figure 89???
References: the presentation of the cited articles is so disastrous that we cannot list all the mistakes and inaccuracies. I have gave up. I recommend that the authors rewrite the citations point by point in accordance with the journal's recommendations. The format must be uniform, and here it is difficult to talk about any format.
References: About 33% of articles/books cited in this manuscript are self-citation of one of the authors. The percent should be lower.
Author Response
Reviewer #1
The reviewed manuscript presents the adsorption kinetic of SARS-Cov-2 spike protein subunit with the receptor binding domain at abiotic surfaces. The manuscript presents very interesting and worth reading results and in my opinion is in the journal’s fields of interests. However, I have same reservations thus my suggestion is major revision.
All manuscript: The manuscript is very carelessly prepared, which is surprising to me because the work was prepared by as many as seven authors and two of them are correspondent authors. Unfortunately, this significantly spoils the good impression of this work. It is good practice to read the final manuscript before submitting it to the editor. Some examples:
Authors’ general comment: We’d like to underline that many of the technical errors were due to the formatting of our manuscript done by the Journal, whereas our original submission was in a proper WORD format. In consequence, the response to the Reviewer comments is only given in places where it was our own fault. Moreover, in order to facilitate further reviewing we have ourselves formatted the manuscript after introducing the corrections suggested by the Reviewer.
lines 4, 5: names/initials and surnames of authors.
Authors’ comment: these corrections suggested by the Reviewer have been introduced
lines 26-27: What is the difference between corona of S protein and S protein corona?
Authors’ comment: in the first key word the stress is on the S protein and in the second on the corona.
line 40: Molecular Dynamics – capital letters?
line 91: two parenthesis?
Fig. 4: why is the signature of the ordinates once horizontally and once vertically?
Fig. 4. Unusual style of legend and figure caption. Please change them into the more conventional way.
lines 327-328: why 3rd but 11th (not 11th)?
line 425: Figure 89???
Authors’ comment: these corrections suggested by the Reviewer have been introduced
How can you explain only the slight dependence of the silica/150 mM NaCl system on changing pH (Fig. 4a)?
Authors’ comment: this slight dependence on pH is probably due to the specific adsorption of Na+ cations in the pores of the silica layer, whereas for molecularly smooth mica (curve 3 in Fig. 5) the adsorption is significantly smaller.
Fig. 5: Is it possible to present the AFM data in the more convincing way? It is really hard to analyze them.
Authors’ comment: we have added the surface concentration of particles in Fig. 5 that facilitates the analysis.
References: the presentation of the cited articles is so disastrous that we cannot list all the mistakes and inaccuracies. I have gave up. I recommend that the authors rewrite the citations point by point in accordance with the journal's recommendations. The format must be uniform, and here it is difficult to talk about any format.
Authors’ comment: the reference list has been rewritten and prepared in a proper format required by the Journal.
References: About 33% of articles/books cited in this manuscript are self-citation of one of the authors. The percent should be lower.
Authors’ comment: the number of self-citations has been reduced.
Reviewer 2 Report
Authors report the study of the binding of the commercial SARS-Cov2 spike protein at abiotic surfaces (i.e. silica, mica and negatively charged polystyrene particles). Their results can be grouped into two categories: one devoted to theoretical estimations of parameters of the S1R subunit (section 3.1) and a second one with experimental data consisting of the characterization of i) the of the surfaces (section 3.2 and 3.3) and ii) the corona formation at sulfonated polystyrene nanoparticles (section 3.4). The manuscript contains some inconsistencies that should be addressed:
-
Size of the protein: according to the authors, they have used a recombinant S1R protein with a molecular weight of 25 kDa (material and methods, line 89-90), in full agreement with the experimental value reported by the supplier (Raybiotet) that also estates that the protein is His-tagged. However, on page 4 (lines 168-169) and in table 1, the value reported is 22 kDa, remarking “theoretical modeling” (page 4 entry 1). I assume that this value is derived from the sequence and I wonder whether the authors included the His-tag. Since the molecular weight is used for estimating other parameters, it is important to define it properly. Please, comment.
-
Density of the protein: authors used a value of 1.36 g/cm3 and cited ref [29]. However, according to refence [29], a more realistic value derived from the analysis of data reported in the bibliography is larger than 1.4 mg/cm3, increasing as the molecular weight of the protein becomes lighter. Please, justify the use of 1.36 g/cm3 (in fact, the classical general value is 1.35 g/cm3, that is used in ref [31]). The value of this parameter is important because it is employed to estimate the hydrodynamic diameter, which is used to discuss the size of the aggregates (section 3.3).
-
propKa is an empirical althorithm that predicts the pKa and charge of the molecule. It is based on interactions among residues and requires the coordinates of the structure as an input file. Did author considered the Hig-tag?
-
In practical terms for this paper, what is the difference between RBD and RBM and what is the aim of figure 2?. I undestrand that figure 2c is intended to demonstrate that the hydrodynamic diameter covers the RBD domain, although if the authors assume that the orientation of S1R at the surfaces is random, it may be interesting to show the hydrodymica diameter for different orientations. However, authors should be aware that the presence of the his-tag (a positive patch) may play a critical role in the interaction with the substrate (negatively charged), favoring certain orientations of the molecule. I strongly suggest analyzing the position of the His-tag (according to the manufacturer it is at the N-end) and taking it into account for the discussion the oligomerization.
-
In figure 3, how RBD and RBM are defined and their coordinates extracted? The statement “As can be seen, a close resemblance for pH range 3.5 to 7.5 is evident, which suggests that myoglobin can be used as an inexpensive model protein for investigating of the S1R adsorption on various surfaces [31].” (page 6 lines 190-193) is not supported by figure 3 (please, compare your figure 3 with figure 4 from ref [31]). Additionally, what is the point of mentioning myoglobin in the context of this manuscript?
-
In table 2, please, define CH (it has the numerical value of the density) and how dH is calculated (the formula depicted as footnote does not fit with the results (for the monomer dH should be CH*d1 and according to table 2 it is d1). Additionally, dAMF for the monomer is 3.72 nm. Is it an experimental value? According to the histogram plotted in figure 6, the smallest value is larger than 5 nm.
-
Authors claim that their results “may be exploited for investigations of the RDB domain interaction with various macromolecule ligands, particularly with immunoglobulins” (page 18, lines 489-491) Do authors have any evidence of the effect of the pH on the stability of the protein? Some adsorption experiments are carried out at pH 3.5 and most proteins are affected. Please, comment. DLS measurement of the protein at different pH may provide additional insight.
-
The solutions at different pH values are isothinic?
Some minor issues. Please, check:
- Letter size on page 2, lines 91-92
- Page 6, line 219 is cut
- Line 6, lines 187-188 repetition
- Letter size in page 11 lines 306-307
- Page 13, line 352 is cut
- Some references do not include the DOI (for example references 2, 14, 21, 31, 37, 38, 39). Please, double check the format.
Author Response
Reviewer #2
Authors report the study of the binding of the commercial SARS-Cov2 spike protein at abiotic surfaces (i.e. silica, mica, and negatively charged polystyrene particles). Their results can be grouped into two categories: one devoted to theoretical estimations of parameters of the S1R subunit (section 3.1) and a second one with experimental data consisting of the characterization of i) the of the surfaces (section 3.2 and 3.3) and ii) the corona formation at sulfonated polystyrene nanoparticles (section 3.4). The manuscript contains some inconsistencies that should be addressed:
- Size of the protein: according to the authors, they have used a recombinant S1R protein with a molecular weight of 25 kDa (material and methods, line 89-90), in full agreement with the experimental value reported by the supplier (Raybiotech) that also estates that the protein is His-tagged. However, on page 4 (lines 168-169) and in table 1, the value reported is 22 kDa, remarking “theoretical modeling” (page 4 entry 1). I assume that this value is derived from the sequence and I wonder whether the authors included the His-tag. Since the molecular weight is used for estimating other parameters, it is important to define it properly. Please, comment.
- Density of the protein: authors used a value of 1.36 g/cm3and cited ref [29]. However, according to reference [29], a more realistic value derived from the analysis of data reported in the bibliography is larger than 1.4 mg/cm3, increasing as the molecular weight of the protein becomes lighter. Please, justify the use of 1.36 g/cm3 (in fact, the classical general value is 1.35 g/cm3 that is used in ref [31]). The value of this parameter is important because it is employed to estimate the hydrodynamic diameter, which is used to discuss the size of the aggregates (section 3.3).
Authors’ comment: These are valid remarks. Indeed, previous calculations were done for the native S1R, without the His-tag, with the molar mass calculated from the amino-acid sequence equal to 25 kDa (this is a corrected value, whereas the previous was improperly calculated). Therefore, according to the Reviewer suggestion we have also calculated the corresponding parameters for the His-tagged molecule with a molar mass of 25.8 kDa and the density of 1.4 g/cm3. These results are collected in Table 1. As can be seen the difference in the equivalent sphere diameters (3.84 and 3.88 nm, respectively) amounts to a fraction of and Angstrom and is, therefore, insignificant given that the AFM resolution is of the order of 0.5 nm at the best.
- PropKa is an empirical althorithm that predicts the pKa and charge of the molecule. It is based on interactions among residues and requires the coordinates of the structure as an input file. Did author considered the Hig-tag?
Authors’ comment: According to the Reviewer suggestion we have performed additional calculations for the His-tagged molecule. The dependence of the nominal charge on pH for this molecule is shown in Figure 3 and compared with the native RBD molecule. As can be expected, the nominal charge of His-tagged molecule is slightly higher in a wide pH range, comparing to native protein. However, taking into account that nominal charge is significantly compensated in electrolyte solution, due to the counterion condensation on oppositely charge amino-acids, the difference in effective charge between native and His-tagged proteins should be negligible.
- In practical terms for this paper, what is the difference between RBD and RBM and what is the aim of figure 2? I understand that figure 2c is intended to demonstrate that the hydrodynamic diameter covers the RBD domain, although if the authors assume that the orientation of S1R at the surfaces is random, it may be interesting to show the hydrodymica diameter for different orientations. However, authors should be aware that the presence of the his-tag (a positive patch) may play a critical role in the interaction with the substrate (negatively charged), favoring certain orientations of the molecule. I strongly suggest analyzing the position of the His-tag (according to the manufacturer it is at the N-end) and taking it into account for the discussion the oligomerization.
Authors’ comment: RBM only represents a small fraction of the RBD molecule, as shown in Figure 1, which is responsible for binding ACE2. In Figure 2c rather than the hydrodynamic diameter we have shown the equivalent sphere diameter, which corresponds to the diameter of the sphere which has the same mass and volume as the RBD molecule. In Figure 2a two different orientations of the RBD are shown. As discussed above the presence of the His-tag slightly affect the nominal charge of the RBD molecule. Perhaps it can locally increase the positive charge but one should remember that it is considerably compensated in electrolyte solutions, typically by a factor of 80% over the distance of ca. 1 nm. In consequence one can expect that the His-tag presence should not influence the interactions among the RBD molecules and the interaction of oligomers with surfaces. Indeed, the presence of His-tag cannot be neglected when considering adsorption of single, non-aggregated protein on the negatively charged surface. However, these issues are beyond the scope of our work experimental work dealing with investigations of the oligomer adsorption kinetics.
- In figure 3, how RBD and RBM are defined and their coordinates extracted?
Authors’ comment: RBD and RBM fragments are defined based amino-acid (aa) sequence provided in the literature (see e.g., Chaki et al, Microbiology Spectrum 10, 2022, e00665-22, RBD (319 to 541 aa) and RBM (438 to 506 aa). The coordinates of the fragments were extracted from the sequence using Visual Molecular Dynamic (VMD) software.
The statement “As can be seen, a close resemblance for pH range 3.5 to 7.5 is evident, which suggests that myoglobin can be used as an inexpensive model protein for investigating of the S1R adsorption on various surfaces [31].” (page 6 lines 190-193) is not supported by figure 3 (please, compare your figure 3 with figure 4 from ref [31]). Additionally, what is the point of mentioning myoglobin in the context of this manuscript?
Authors’ comment: These is a valid remark. The myoglobin curve has been deleted from Figure 3 as well as the corresponding discussion in the text.
- In table 2, please, define CH (it has the numerical value of the density) and how dH is calculated (the formula depicted as footnote does not fit with the results (for the monomer dH should be CH*d1 and according to table 2 it is d1). Additionally, dAMF for the monomer is 3.72 nm. Is it an experimental value? According to the histogram plotted in figure 6, the smallest value is larger than 5 nm.
Authors’ comment: It should be mentioned the formula for the dH calculation is only valid for the aggregation degree na larger than 10, where the oligomers assume a quasi-spherical shape. This statement has been added to the legend of table 2. The value of d1 for the monomer corresponds to the diameter of the equivalent previously calculated and given in Table 1. As far as the histogram is concerned one can expect that the number of monomers on the surface is very low because of their low concentration in the suspension. Additionally, because of their small size they are less visible by AFM due to the tip convolutions effect.
- Authors claim that their results “may be exploited for investigations of the RDB domain interaction with various macromolecule ligands, particularly with immunoglobulins” (page 18, lines 489-491) Do authors have any evidence of the effect of the pH on the stability of the protein? Some adsorption experiments are carried out at pH 3.5 and most proteins are affected. Please, comment. DLS measurement of the protein at different pH may provide additional insight.
Authors’ comment: It this statement we meant the RBD oligomer corona immobilized at the polymer particles. The advantage of this system is that the corona can be effectively produced only using microgram quantities of the protein. In contrast the protein zeta potential, size and stability as a function of pH using DLS require at least 100 to 1000 larger protein quantities, which is impractical given the high price of the RBD protein. One should also underline that the particles with the corona are stable over a wide pH range comprising 7.4 that enables the interactions with the immunoglobulins to be determined under physiological conditions.
The solutions at different pH values are isothinic?
Authors’ comment: yes, the solutions at different pHs were kept at the fixed ionic strength, either 10 or 150 mM.
Some minor issues. Please, check:
- Letter size on page 2, lines 91-92
- Page 6, line 219 is cut
- Line 6, lines 187-188 repetition
- Letter size in page 11 lines 306-307
- Page 13, line 352 is cut
Authors’ comment: These technical errors were due to the formatting of our manuscript done by the Journal, whereas our original submission was in a proper WORD format. In order to facilitate further reviewing we have ourselves formatted the manuscript after introducing the corrections correction suggested by the Reviewer.
- Some references do not include the DOI (for example references 2, 14, 21, 31, 37, 38, 39). Please, double-check the format.
Authors’ comment: the reference list has been rewritten and prepared in the proper format required by the Journal.
Round 2
Reviewer 1 Report
Manuscript can be accepted for publication.
Author Response
Thanks, we appreciate your positive review.
Reviewer 2 Report
The authors have successfully addressed most of the comments by this referee. However, two issues deserve additional attention:
-
The calculation of dH in Table2 is not fully clear. Please, define CH and explain how you estimate dH for na<10. This is important because you use the values of dH to estimate the size of the aggregates on the surface of the material detected by AFM.
-
Authors minimize the influence of the his-tag (please, in line 284 replace 3.8 nm by 3.9 nm). Recall that the purification of his-tagged proteins is based on the interaction of the his-tag with a solid support. Hence, although charge screening occurs under physiological conditions, the presence of 6 contiguous His residues is very relevant in the interaction with negatively charged supports. These 6 His residues play a major role in both orientation and strength of the interaction with the support. Since the position of the His-tag is known (N-end), I strongly suggest a discussion on how it may favor certain orientations. This is important if, as commented in lines 534-536, “Such layers can be exploited for investigations of the protein interactions with various abiotic surfaces and macromolecule ligands, particularly with immunoglobulins”.
Additionally
a. For same readers, the use of S1R and RBD may be confusing. I would suggest using only S1R
b. Please, double check the bibliogragraphy. Some information is missing. For example, in Ref. 2, page numbers are missing (they have been included as part of the DOI and in ref 12, the last page is missing.
Author Response
Reviewer #2
- The calculation of dH in Table2 is not fully clear. Please, define CH and explain how you estimate dH for na<10. This is important because you use the values of dH to estimate the size of the aggregates on the surface of the material detected by AFM.
Authors’ comment: For the aggregate composed of two and four molecules in regular and hexagonal arrangement the hydrodynamic diameters were calculated in an exact way by applying the numerical multipole-expansion method (Refs. 33, 40).
On the other hand, for na> 10 the hydrodynamic diameter was calculated from the formula, which was derived assuming a quasi-spherical shape of the aggregates. The value of the constant was estimated assuming a given porosity of the aggregate. It should be mentioned that this formula gives only a first-order estimate of the hydrodynamic diameter, more exact calculations require considerable numerical effort and were not performed.
- Authors minimize the influence of the his-tag (please, in line 284 replace 3.8 nm by 3.9 nm). Recall that the purification of his-tagged proteins is based on the interaction of the his-tag with a solid support. Hence, although charge screening occurs under physiological conditions, the presence of 6 contiguous His residues is very relevant in the interaction with negatively charged supports. These 6 His residues play a major role in both orientation and strength of the interaction with the support. Since the position of the His-tag is known (N-end), I strongly suggest a discussion on how it may favor certain orientations. This is important if, as commented in lines 534-536, “Such layers can be exploited for investigations of the protein interactions with various abiotic surfaces and macromolecule ligands, particularly with immunoglobulins”.
Authors’ comment: This is an interesting remark. Indeed, for the S1R-His-taged molecule in a monomeric state the interactions with the support (in a chromatographic column) can play a significant role despite that the effective charge is only increased by a fraction of an elementary charge as a result of ion condensation and electrostatic screening. This is so because in the column the molecules make a considerable number of contacts with the substrate, which slow their propagation but do not lead to irreversible adsorption. In this case, the small differences in the interaction energy can create large effects.
However, in our experiments, the situation was significantly different because the molecules formed aggregates in which the location of the His-tag motif cannot be a priori determined. Additionally, the adsorption mechanism of such aggregates at flat substrates is essentially different in comparison to the chromatographic column because they irreversibly adsorb upon contacting the surface. Under such mechanism, the interaction energy (especially the van der Waals contribution) stems from a considerable part of the molecule with the His-tag contribution playing a less important role.
Because of the significance of the His-tag issue for the S1R molecule stability and interactions with surfaces, thorough Molecular Dynamics modeling for monomeric molecules is planned in our future works. At this stage, however, we are not able to define the SR1 protein orientation on the substrate. Therefore, to avoid confusion, we have modified the sentence mentioned by the Reviewer.
- For some readers, the use of S1R and RBD may be confusing. I would suggest using only S1R.
Authors’ comment: This is a valid remark, we have only used S1R throughout the manuscript as suggested by the Reviewer.
- Please, double-check the bibliography. Some information is missing. For example, in Ref. 2, page numbers are missing (they have been included as part of the DOI and in ref 12, the last page is missing.
Authors’ comment: the reference list has been double-checked and corrected as suggested by the Reviewer.